# Topic recognition and refined evolution path analysis of literature in the field of cybersecurity

**Yanfeng Zhu[1], Zheng Li[2], Tianyi Li[1]\*, Lei Jiang[2]**

1 School of Computer and Big Data, Heilongjiang University, Harbin, China, 2 School of Information Management, Heilongjiang University, Harbin, China

\* litianyi@hlju.edu.cn

## Abstract

Using text analysis techniques to identify the research topics of the literature in the field of cybersecurity allows us to sort out the evolution of their research topics and reveal their evolution trends. The paper takes the literature from the Web of Science in the field of cybersecurity research from 2003 to 2022 as its research subject, dividing it into ten stages. It then integrates LDA and Word2vec methods for topic recognition and topic evolution analysis. The combined LDA2vec model can better reflect the correlation and evolution patterns between adjacent stage topics, thereby accurately identifying topic features and constructing topic evolution paths. Furthermore, to comprehensively evaluate the effectiveness of the LDA model in topic evolution analysis, this paper introduces the Dynamic Topic Model (DTM) for comparative analysis. The results indicate that the LDA model demonstrates higher applicability and clarity in topic extraction and evolution path depiction. In the aspect of topic content evolution, research topics within the field of cybersecurity exhibit characteristics of complexity and diversity, with some topics even displaying notable instances of backtracking. Meanwhile, within the realm of cybersecurity, there exists a dynamic equilibrium between technological developments and security threats.

## 1. Introduction

In recent years, with the proliferation of cybersecurity incidents worldwide, the issue of cybersecurity has emerged as a hot topic of global concern. These cybersecurity incidents not only pose significant challenges to individual privacy and corporate operations but also present severe threats to national security. Consequently, research in the field of cybersecurity holds paramount importance. This study aims to delve into the evolutionary path of research literature in cybersecurity, with a particular emphasis on the research topics within the technological domain and their historical evolution. Through an in-depth analysis and excavation of a vast corpus of literature, this paper elucidates the core elements and potential threats inherent in cybersecurity issues, thereby providing theoretical underpinnings for the formulation of effective defensive strategies. Moreover, this study employs text mining techniques to conduct systematic topic identification and evolutionary path analysis of literature within the cybersecurity domain. The objective is to construct a comprehensive and systematic knowledge

**Data availability statement:** https://figshare.com/articles/dataset/data/28089812.

**Funding:** This study is funded by the Project of Social Science Foundation of Heilongjiang Province (Number: 23TQD174).

**Competing interests:** No authors have competing interests.

framework that can robustly support further research endeavors in the field of cybersecurity. Research in the field of cybersecurity not only holds significant practical relevance for safeguarding national cyberspace security and protecting the interests of citizens and enterprises but also carries profound strategic value for promoting harmony, security, and sustainable development in the global cyberspace. This research enables a deeper understanding of the latest developments and technological trends within the cybersecurity domain, thereby providing valuable reference information for policymakers, researchers, and practitioners. Furthermore, this paper will facilitate the inheritance and innovation of knowledge within the cybersecurity field, propelling the domain towards higher levels and broader horizons of development.

Currently, research in the field of cybersecurity primarily focuses on specific technological implementations and applications, with a lack of systematic studies on the historical evolution of research topics within this domain. Developmental research mainly emphasizes standards and strategic advancements in specific regions. In exploring of the overall development of the cybersecurity field, scholars predominantly employ three categories of methodologies for in-depth investigation. The first category involves an investigative and summarizing approach, where scholars conduct comprehensive analyses and summaries of the status, issues, and trends in the field of cybersecurity by collecting extensive empirical data and case studies. For instance, Li Y et al. [1] provide a comprehensive review the progress of standards in the field of cybersecurity through systematic surveys and summaries. They offer an exhaustive introduction to new trends, latest developments, security threats, and challenges within the domain. Kaur J et al. [2] also employed an investigative and summarizing approach to provide a comprehensive introduction to various existing advanced cybersecurity standards and employed the challenge faced by the field of cybersecurity. The second category is based on traditional bibliometric methods. Scholars primarily employed conventional approaches such as scientometrics and bibliometrics to identify research hotspots and developmental trends with using in the cybersecurity field. For instance, Matilde-Espino Y et al. [3] conducted a bibliometric analysis of scientific articles published between 2015 and 2022 concerning cybersecurity issues in Mexico. Elango B et al. [4], utilizing the Scopus database and employing scientometric methods, conducted an in-depth analysis of cybersecurity research by Indian authors over the past three years. They not only identified the major trends and patterns within these studies but also provided a detailed description of the transformations and developments in these research topics. Sharma D et al. [5] employed citation analysis and other methods to conduct an extensive bibliometric analysis of literature in the field of cybersecurity published in the Web of Science database between 2011 and 2021. Their research unveiled the developmental trends of significant topics within the cybersecurity domain over the past decade, as well as their citation patterns. Daim T et al. [6] utilized scientometric methods to analyze the research on cybersecurity issues in journal articles. Although the aforementioned two methods can quickly identify highly credible research content, they fail to account for relationships between synonyms, stop words, and latent semantics. Not only are they unable to accurately extract topics, but they are also prone to overlooking potential research hotspots. The final category involves topic identification methods based on text mining. These approaches primarily employ natural language processing techniques to extract and categorize potential topics from extensive textual data. The most commonly used topic model is the Latent Dirichlet Allocation (LDA) model. By applying these topic models, scholars can effectively uncover research hotspots within the field of cybersecurity, thus obtaining a comprehensive overview of research in this domain. For instance, Song M et al. [7] employed the LDA topic model to conduct an in-depth quantitative analysis of cybersecurity strategies in countries such as the United States, the United Kingdom, and Japan. This research

provided valuable references and recommendations for the revision of future National Cyber Security Strategies (NCSS) in South Korea. Hwang S Y et al. [8] utilized the LDA method to extract key technological topics from a vast number of academic papers and patent literature, thereby identifying the security integration domains and technological development trends within the field of cybersecurity. Khandelwal S et al. [9] treated tweets related to cybersecurity on social media platforms as research subjects, employing the LDA model to conduct an in-depth analysis of public discussions and perceptions regarding cybersecurity issues during the COVID-19 pandemic.

In 2003, Blei et al. [10] introduced the Dirichlet distribution and proposed the Latent Dirichlet Allocation (LDA) model for the first time. The LDA model is an unsupervised learning technique that can be used to recognize latent topic information in large-scale document collections [11] and is often employed to mine latent topic information in corpora within big data environments. For instance, Buenano-Fernandez et al. [12] utilized the LDA topic model to mine self-assessments of university teachers, extracting valuable topic information and thus assisting researchers in decision-making. With the advancement of machine learning technologies and the continuous maturation of LDA models in application, an increasing number of scholars have begun to investigate the trends of topic evolution based on LDA. These scholars use the LDA model for topic extraction and incorporate variables such as time dimensions into the model. This allows them to capture the dynamic changes of topics over time and subsequently discuss the research frontiers and evolution trends in a specific academic field [13]. For instance, Zhao et al. [14] utilized the LDA model to extract topics and performed time slicing on the identified topics, extracting the topic and topic word distribution for each time slice, thereby revealing the evolution trends of the topics. Wu H et al. [15] proposed a comprehensive method that combines LDA-based topic recognition analysis, improved topic lifecycle analysis, and enhanced technology entropy analysis. This method can be used to identify, measure, and interpret the evolution of topics within patent documents. Jiang et al. [16] constructed a product online review topic evolution analysis model based on the LDA model, tailored to the characteristics of product forums. This model can mine the regularities of topic evolution of online product forums from three aspects: topic labels, topic popularity, and topic word popularity. However, the LDA model tends to overlook the issue of potential semantic associations between text contexts. In the field of natural language processing, deep learning models such as Word2vec can effectively convert unstructured text data into semantically rich word vectors. This model significantly improves the performance of contextual logical relationship recognition and similarity measurement. For instance, Xi et al. [17] studied a technology similarity visualization method based on Word2vec and LDA topic models. They conducted an empirical study using the NEED domain as a case study and verified that the method showed good results in technology similarity analysis. Therefore, the combined LDA2vec model can better reflect the relationships and evolution patterns between topics in adjacent stages. This allows for more accurate identification of topical features and the construction of topic evolution paths.

In addition, some scholars have begun to employ the Dynamic Topic Model (DTM) to research the evolution of topics. For example, in 2012, Li D et al. [18] conducted experiments that demonstrated the DTM's ability to dynamically process time-series document datasets. The DTM represents an improvement and expansion over the LDA model. The DTM enables the identification and tracking of dynamic topics in these datasets and can reveal coordinated evolution patterns of topics and topic words in a specific field. The DTM demonstrates superior topic detection capabilities compared to textual methods such as word frequency and co-word analysis [19]. The underlying principle of its application is as follows: first, discretize the retrieved literature abstracts into time-ordered slices. Subsequently, it is assumed that both

the topic distribution and content in neighboring time slices evolve over time. Ultimately, the chain of topics across a time-continuous set of abstract data is identified [20]. For example, Wu et al. [21] proposed a method for topic evolution that integrates the Dynamic Topic Model (DTM) with community detection techniques. This method can be used to address the issues that traditional dynamic topic models encounter with high-dimensional and sparse data. Shi L et al. [22] focused on the sparsity issue in datasets and the dynamic changes of topics, using the DTM algorithm to capture topic distributions and evolutions. The experiment indicated that this algorithm's topic coherence and clustering quality are superior to those of some other methods.

In summary, although significant research progress has been made in the field of cybersecurity, current studies predominantly rely on traditional bibliometric methods, with relatively fewer applications of topic model-based approaches. Furthermore, most studies have a short duration, lacking in-depth analyses of long-term development trends. Considering this, to systematically understand the research landscape in this field, this paper employs topic modeling techniques to conduct a comprehensive analysis of the literature in the cybersecurity domain over the past two decades. This approach aims to fully reveal the research dynamics and evolutionary trajectories within this field. Additionally, during the process of topic evolution analysis, this study also conducts a comparative analysis between LDA2vec and DTM models to evaluate the effectiveness of both models in the research process. Finally, this study constructs a visualized Sankey diagram to intuitively present the evolutionary paths of research topics. This research focuses on addressing the following two core issues:

Question 1: What methods or technical frameworks should be employed to achieve a refined analysis of the topic evolution paths?

Question 2: In the field of cybersecurity, what characteristics and patterns are exhibited by the refined evolution paths of topics?

## 2. Research framework

To gain an in-depth understanding of the research dynamics and future developmental trends in the field of cybersecurity, this study is primarily divided into four steps: data acquisition and preprocessing, topic model evaluation, topic identification, and refined evolution path analysis. The overall analysis framework is illustrated in Fig 1. Data acquisition and preprocessing form the foundation of the entire research, involving the collection of literature data, as well as format processing and data cleaning tasks. In the section on evaluating topic models, this study primarily conducts a comparative analysis between the LDA2vec model and the DTM model from two aspects: the acquisition of topic words and the scoring of topic coherence. Topic identification will apply the evaluated topic models to analyze the preprocessed literature data, aiming to identify the primary topics within the literature. Finally, based on the identified topics, this study conducts an evolutionary path analysis to reveal the trends of these topics over time.

### 2.1. Data acquisition and preprocessing

**2.1.1. Data acquisition.** Since the beginning of this century, the core focus of network construction and development has been on the establishment and maintenance of infrastructure. However, with the continuous growth in the number of Internet users, cyber-attack events have become increasingly frequent, prompting governments and enterprises to gradually recognize the importance of cybersecurity. Since 2003, an increasing number of scholars have shifted their research focus toward the field of cybersecurity.

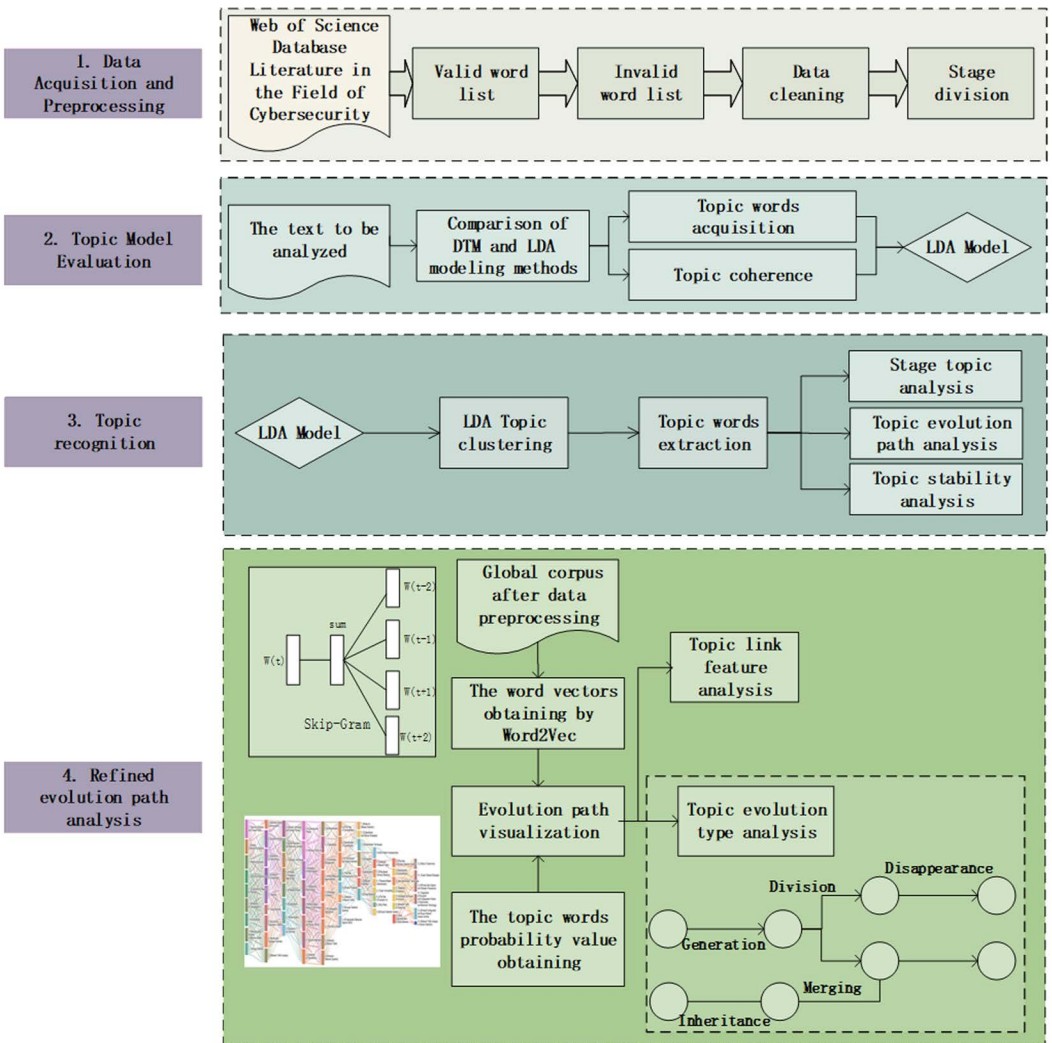

**Fig 1. Overall analysis framework diagram.**

In text mining research, sample size is crucial for ensuring the reliability and validity of results. To guarantee the representativeness and quality of the analysis, this study retrieved literature related to network security from the "Web of Science" database, covering the period from 2003 to 2022, with the retrieval conducted in October 2023. This timeframe encompasses the rapid development phase of the network security field. Using the search query "TS=('network security' OR 'cybersecurity' OR 'cyber security' OR 'cyber-security')", a total of 52,695 articles were obtained. To enhance the quality of the analysis, articles lacking titles, keywords, or abstracts were excluded, resulting in a final sample of 50,657 articles for topic modeling analysis. This sample size, while maintaining data quality and completeness, provides sufficient literary support to ensure the accuracy and effectiveness of the analysis results.

Ultimately, after screening, this study identified 50657 articles as the sample for topic modeling analysis. This sample size reflects the sufficient volume of literature required by researchers to support the validity of analyses and conclusions; while maintaining data quality and informational integrity.

**2.1.2. Data preprocessing.** Data preprocessing is a crucial step in text analysis, primarily encompassing methods such as data cleaning, data integration, data transformation, and data reduction [23]. Considering the linguistic characteristics of literature data, this study conducted preprocessing operations on the corpus, which primarily included the following four steps:

(1) Valid word list construction: Considering the rich knowledge of the target domain, the unique professional vocabulary, and the real challenges of unsupervised learning [24], this paper acquires the subject literature through the Web of Science. We then extract the titles, keywords, and abstracts to construct a valid word list.

(2) Invalid word list construction: To improve the accuracy of topic recognition for the target domain and avoid the interference of high-frequency invalid words in various fields [24], this paper proposes a method to construct a domain-specific invalid word list based on a general stop word list. First, we repeatedly cluster and filter high-frequency keywords with no practical meaning using the LDA model. Subsequently, these words are extracted into a general stop word list, finally forming the final word list.

(3) Data cleaning. To minimize noise and facilitate easier analysis and handling of textual data, this study conducted data cleaning on the literature dataset. Firstly, special elements such as numbers and punctuation were removed. Subsequently, the terms were converted to singular form and lowercase. Finally, Python was utilized to perform operations on the text such as segmentation, loading a valid word list, and removing invalid words.

(4) Stage division. After preprocessing the collected literature, this study divided the documents into ten stages, each spanning two years. This was done to present the development and changes in the field of cybersecurity in a more systematic and organized manner.

## 2.2. Topic model evaluation

Topic evolution models are used to capture the dynamic characteristics of topics over time. Commonly used topic evolution models include Dynamic Topic Models (DTM), LDA2vec, Deep Learning Models for Topic Evolution, Temporal LDA, and Graph-Based Topic Evolution Models, among others. Among them, DTM is particularly suitable for topic evolution analysis that requires precise modeling of the temporal dimension, but it has greater complexity and computational overhead. The LDA2vec model possesses better semantic understanding capabilities and is suitable for processing dense text data, but it is relatively weak in terms of dynamic temporal analysis. Therefore, this paper divides the time series into stages and then conducts topic analysis based on the LDA2vec model, attempting to address its dynamic time analysis issues through stage division. The study fully leverages the semantic understanding capability of LDA2vec to complete the thematic evolution analysis in the field of cybersecurity. By comparing it with the thematic evolution analysis of the DTM model, the paper aims to select the optimal model for the thematic evolution of research literature in the field of network security.

To systematically analyze the literature in the field of cybersecurity from 2003 to 2022, this paper employed both DTM and LDA2vec models to explore the evolution of topics within the literature. In this study, the number of literature from the early years is limited. Therefore, we divided the literature into stages every two years, resulting in a total of 10 stages. This approach provides balanced and sufficient data, thereby optimizing the effectiveness of the topic analysis. At the same time, due to the rapid changes in research focus and technological developments within the field of cybersecurity, relevant literature often responds swiftly

to emerging threats, technologies, and policies. The biennial stage division method not only helps capture immediate changes but also ensures the coherence and consistency of literature topics within each stage, providing support for in-depth topic evolution analysis. Finally, these ten stages provide us with a time series framework to track the evolution of cybersecurity research. After a detailed comparative analysis of DTM and LDA, we found that the two models exhibited significant differences in terms of topic word acquisition and topic coherence.

(1) DTM model

As shown in Fig 2, the DTM model does not perform well in distinguishing topic words. This indicates that the model has certain limitations in identifying and distinguishing various types of topic words. These limitations may stem from the DTM model's inadequacies in handling complex text structures, making it challenging to capture the semantic patterns of topics within the text [25].

The topic coherence score is a crucial metric for evaluating the quality of topic models, assessing the coherence and interpretability of the generated topics. A higher coherence score indicates that the words within a topic are more semantically related, enhancing the topic's interpretability. Conversely, a lower score suggests weaker semantic associations among words, making the topic harder to comprehend. Studies have demonstrated that topic coherence scores align closely with human understanding [26]. Typically, a higher topic coherence score signifies a more optimal selection of the number of topics. In a systematic analysis of cybersecurity literature from 2003 to 2022, both the Dynamic Topic Model (DTM) and Latent Dirichlet Allocation (LDA) were employed to explore topic evolution. The literature was divided into ten biennial phases, providing a temporal framework to trace the evolution of cybersecurity research. Fig 3 presents the topic coherence scores of DTM and LDA across these ten-time windows. A detailed comparative analysis of DTM and LDA revealed significant differences in topic coherence. Although DTM's topic coherence scores exhibit a gradual upward trend over time and surpass those of LDA in most phases, its effectiveness in elucidating topic evolution paths is limited. This limitation primarily stems from DTM's insufficient flexibility and semantic depth in handling cross-period topic evolution, hindering its ability to accurately capture the complex dynamics of cybersecurity research over time [25]. To address DTM's shortcomings, this study integrates the strengths of LDA and Word2Vec, linking topic distributions with word vector spaces. This approach leverages both statistical and semantic information in topic extraction, enhancing the performance and interpretability of the topic model. However, LDA is not inherently designed to model the temporal evolution of topics. Segmenting the literature into temporal phases mitigates this issue to some extent, but significant variations in data volume and topic distribution across phases lead to fluctuations in topic coherence scores. The coherence scores were calculated using cosine similarity [27].

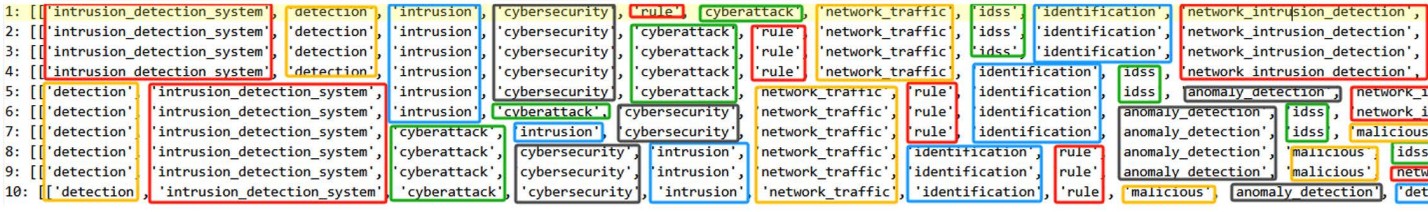

**Fig 2. DTM topic words.**

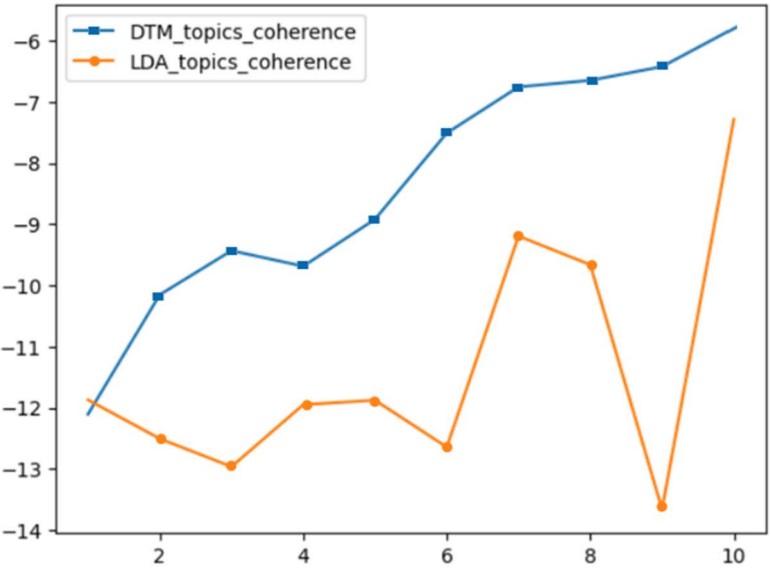

**Fig 3. Analysis of topic coherence.**

(2) LDA2vec

In comparison, the LDA2vec model demonstrates greater topic differentiation (as shown in Table 1 below) and the capability to analyze evolution paths. Through the training of word vectors using the Word2vec model, the LDA method is not only able to effectively capture the deep semantic connections between topics but also accurately track the evolution of topics between different time stages [28]. LDA2vec can more comprehensively reflect the development trends and changes in the field of cybersecurity research, providing researchers with a deeper and more systematic analytical perspective.

In summary, the DTM model provides an intuitive framework for analyzing how topics change over time. However, based on the application performance in this study, LDA2vec exhibits superior performance in the analysis of topic evolution in cybersecurity literature. Therefore, LDA2vec was selected as the primary analytical tool for this study due to its superior semantic processing capability and topic recognition accuracy. However, the DTM model still has its unique advantages in analyzing how topics evolve, and future research could explore combining the strengths of both models to achieve more comprehensive and in-depth text-mining analysis results.

## 2.3. Topic recognition

Before starting the training of the LDA model, this paper determines the number of topics K by using the perplexity evaluation method. In general, when the downward trend of perplexity stops or is at an inflection point, the perplexity value K corresponding to this point is the optimal number of topics [29]. After determining the optimal number of topics for each stage, we use the Latent Dirichlet Allocation (LDA) model to perform topic clustering on the literature data of each stage. Subsequently, keywords corresponding to each topic in each stage are extracted. Finally, the analysis is carried out from three aspects: stage topics, topic features, and topic stability.

**Table 1.  Topic words.**

| Stages | Topic words |
|---|---|
| Stage 1 | 1_1(Denial of Service); 1_2(Security Infrastructure); 1_3(Cybersecurity Measures or Practices); 1_4(vulnerability); 1_5(Resolving Issues); 1_6(Latest Cybersecurity Reports); 1_7(Security Protocols and Firewall Rules); 1_8(Detection of Cyberattacks); 1_9(Reporting and Handling of Malicious Nodes); |
| Stage 2 | 2_1(Key Elements); 2_2(Detection of Cyberattacks); 2_3(Blocking Servlet Communication); 2_4(Authorized Passage of Servlets); 2_5(analysis of cyberattacks); 2_6(Listing Information); 2_7(Identity); 2_8(Enhancing Capacities to Defend); 2_9(Network Traffic Analysis); |
| Stage 3 | 3_1(Detection Techniques); 3_2(Policies or Strategies Against DoS); 3_3(Password Rules); 3_4(Identity Verification and Firewall Policies); 3_5(Solutions and Services); 3_6(cyberattack); 3_7(authentication); 3_8(Data Acquisition and Replication Strategies); |
| Stage 4 | 4_1(Establishing Robust Rules); 4_2(Detection of Cyberattacks); 4_3(Roles and Access Control); 4_4(Access Control List); 4_5(Cybersecurity Threats); 4_6(Analysis of Attack Graphs); 4_7(Security Approach); 4_8(Policy-Based Network Traffic Analysis); |
| Stage 5 | 5_1(Cyberattack); 5_2(Detection of Network Traffic); 5_3(Vulnerability Management); 5_4(Policy Measures Against Botnet); 5_5(Strategies to Defend); 5_6(Intrusion Detection Systems); 5_7(The Role of Rules); 5_8(Cryptographic Applications); 5_9(Authentication); |
| Stage 6 | 6_1(Detection of Network Traffic); 6_2(Identification of Cyberattacks); 6_3(Cryptographic Measures Against DDoS); 6_4(Intrusion Detection Systems); 6_5(Privacy Protection); 6_6(The Role of Vulnerabilities); 6_7(The Role and Importance of SSL); 6_8(The Evolution and Impact of Botnets); |
| Stage 7 | 7_1(Authentication Techniques); 7_2(Detection of Botnets); 7_3(Rules for Malware Detection); 7_4(Encryption); 7_5(Identification); 7_6(Threat Detection); 7_7(Cyberattacks and Defense Strategies); |
| Stage 8 | 8_1(Password-Based Authentication); 8_2(Detection of Network Traffic); 8_3(Key Risks); 8_4(The Role of Evaluation in); 8_5(Rule-Based Anomaly Detection); 8_6(DoS Attack Consequences); 8_7(Vulnerability); 8_8(Intrusion Detection System); |
| Stage 9 | 9_1(Role of Authentication in DDoS Defense); 9_2(The Role of Intrusion Detection Systems); 9_3(network traffic & cybercrime); 9_4(Validated Techniques for Intrusion); 9_5(Key Identification Techniques); 9_6(Identification of Vulnerabilities); 9_7(Detection of malware); |
| Stage 10 | 10_1(Cyber Defense Strategies); 10_2(Network Traffic Analysis in Intrusion Detection); 10_3(Role of Cybercrime); 10_4(Cyberattack Identification); 10_5(Firewall Configuration and Secure Network Access Controls); 10_6(Private Data Capture and Utilization Precautions); 10_7(Application of Encryption and Cryptographic Policies in Ransomware and Blockchain Technology); |

## 2.4.  Evolution path analysis

Based on the results of topic recognition, this paper multiplies the LDA probability values corresponding to the topic words of each stage with the word vectors trained by Word2vec. The weighted sum value is then obtained, which results in a vectorized representation of the topics. Subsequently, we calculated the similarity between topics in adjacent stages and determined the topic evolution paths by setting a similarity threshold. Finally, the topic evolution paths were displayed in a visualized form, and the analysis of the evolution paths was conducted from two perspectives: topic linkage characteristics and topic evolution types.

## 3.  Results

### 3.1.  Topic recognition results

In order to identify the core research themes and their evolution across different stages in the field of cybersecurity, this study employed the LDA model for training and topic extraction from the literature. The LDA model analyzes a collection of documents to identify multiple topics, each represented by a set of words and their associated probability distribution. However, LDA does not automatically generate labels for these topics, requiring manual assignment based on the semantic meanings of the high-probability terms within each topic. For instance, in Topic 2 of Stage 2, the high-probability words and their probabilities are as follows: detection (0.073), cyberattack (0.050), cybersecurity_threats (0.033), forms (0.029), survive (0.023), cybersecurity (0.015), identification (0.015), spurious (0.012), avoid (0.011), and protecting (0.011). Based on the semantics of these high-probability words, the topic can be labeled as "Stage 2_2: Detection of Cyberattacks." The topic words listed in Table 1 were generated following this procedure.

Additionally, the number of topics for each stage was determined by calculating the perplexity for each stage. To avoid overfitting, the value corresponding to the inflection point

of the perplexity curve was chosen as the optimal number of topics. Using this method, the optimal number of topics for the ten stages was determined to be 9, 9, 8, 8, 9, 8, 7, 8, 7, and 7, respectively.

**3.1.1. Stage topic analysis.** From the perspective of text mining, there are close connections and co-occurrence relationships between these topics, forming a keyword network in the field of cybersecurity. From the perspective of cybersecurity, these topics encompass the core elements and strategies required to ensure cybersecurity, serving as important support for building a secure and reliable online environment. At the same time, these topics also reflect the complexity and diversity of cybersecurity, which requires a comprehensive application of various technical measures and management strategies to address the ever-evolving threat landscape.

Furthermore, these topics not only mirror the current challenges and best practices within the field of cybersecurity but also indicate the directions and measures that need to be taken to confront future threats. Therefore, it is necessary to conduct a comprehensive summary and analysis of these topic words.

(1) Stage 1 (2003–2004). The first stage focuses on concerns regarding cyberattacks and defense, security infrastructure and protocols, everyday security management, and the latest security reports. These topics collectively form a complete view of the cybersecurity landscape, encompassing aspects ranging from infrastructure construction to daily operational management, as well as attack detection and response.

(2) Stage 2 (2005–2006). The second stage predominantly involves an in-depth discussion of cyberattack detection, analysis, and defense. It also focuses on critical aspects such as servlet communication control and authorization, identity management and information lists, and network traffic analysis.

(3) Stage 3 (2007–2008). Stage 3 provides an in-depth look at a few key aspects of cybersecurity, including attack detection and defense, identity verification and password policies, and cybersecurity solutions and services. Additionally, this stage emphasizes the significance of identity verification technologies in cybersecurity and the application of data acquisition and replication strategies for enhancing cybersecurity. These areas represent the hotspots and trends within the cybersecurity domain at that time, playing a crucial role in protecting sensitive information and ensuring the proper functioning of network systems.

(4) Stage 4 (2009–2010). Stage 4 delves into several critical aspects of the cybersecurity domain, including policies and rules, attack detection and defense, roles and access control, and overall security approaches.

(5) Stage 5 (2011–2012). The topics in Stage 5 cover multiple core areas of the cybersecurity field, including attacks and defenses, network monitoring and intrusion detection, cybersecurity management and rule, botnet countermeasures, as well as cryptography and authentication.

(6) Stage 6 (2013–2014). The sixth stage thoroughly examines several critical facets of the cybersecurity field. These include network traffic detection, identification of cyberattacks, cryptographic defense measures, privacy protection, vulnerability exploitation, the importance of SSL, and the impact of botnets.

(7) Stage 7 (2015–2016). Stage 7 is mainly characterized by in-depth research into encryption and authentication technologies, continuous optimization of threat detection algorithms and tools, and exploration of comprehensive and flexible cybersecurity and defense strategies.

(8) Stage 8 (2017–2018). The topics in Stage Eight primarily include aspects such as authentication and key management, network traffic and intrusion detection, evaluation and anomaly detection, as well as attack consequences and vulnerability management. These topics collectively reflect the latest advancements and areas of focus in the cybersecurity field regarding defense, detection, evaluation, and response. They provide vital guidance and references for building a more secure and reliable cyber environment.

(9) Stage 9 (2019–2020). The ninth stage is predominantly characterized by a comprehensive focus on critical cybersecurity technologies, such as identity authentication, key technologies, intrusion detection and validation, network traffic analysis, and identification of vulnerabilities and detection of malware.

(10) Stage 10 (2021–2022). The topics of Stage 10 are mainly distinguished by intensified research into cybersecurity defense and detection technologies, a focus on issues of cybercrime and data privacy protection, and the exploration of security applications within emerging technologies.

**3.1.2. Topic evolution path analysis.** Based on the topic words from these ten stages, this paper analyzes the development trends in the cybersecurity field from multiple perspectives. This includes the evolution of attack and defense technologies, infrastructure maturation and expansion, identity and access management, data protection and privacy, intrusion detection techniques, and traffic analysis technologies. The proportion of literature in these areas at each stage is shown in Table 2.

The following is a detailed analysis of the development trends in these areas:

(1) Attack and defense techniques

Stages 1 to 3: Research focuses on fundamental defense strategies, covering policies or strategies against Denial of Service (DoS) (stage 1_1) and the enhancement of Cybersecurity Infrastructure (stage 2_1, stage 3_5).

Stages 4 to 6: As attack methods become increasingly sophisticated, the research focus shifts toward more advanced attack detection technologies, such as deep analysis of Network Traffic (stage 4_8) and optimization of Intrusion Detection Systems (IDS) (stage 5_6).

Stages 7 to 10: Research goes further, with scholars beginning to explore complex attack and defense strategies. This includes the widespread application of Encryption Technologies (stage 7_7), the perfection of Privacy Protection Mechanisms (stage 8_1), and the refinement of defense measures against specific types of attacks such as Distributed Denial of Service (DDoS) and Botnets (stage 6_8, stage 9_1).

(2) Infrastructures

Stages 1 to 3: Establishing foundational cybersecurity infrastructure (stage 1_2, stage 1_3), such as Firewall Rules and Security Protocols (stage 1_7).

**Table 2. Distribution of literature proportions by research perspective and stage.**

| Research Perspective | stage 1 | stage 2 | stage 3 | stage 4 | stage 5 | stage 6 | stage 7 | stage 8 | stage 9 | stage 10 |
|---|---|---|---|---|---|---|---|---|---|---|
| Attack defense | 37.62% | 39.61% | 38.01% | 37.26% | 39.23% | 44.85% | 44.30% | 46.07% | 47.60% | 50.52% |
| Intrusion detection | 23.35% | 19.78% | 18.31% | 16.93% | 12.61% | 12.33% | 10.72% | 9.81% | 11.14% | 12.39% |
| Traffic analysis | 1.00% | 1.81% | 1.79% | 1.37% | 1.76% | 2.01% | 1.44% | 1.35% | 1.21% | 1.16% |
| Infrastructures | 0.70% | 0.62% | 0.54% | 0.66% | 0.52% | 0.66% | 0.56% | 0.65% | 0.67% | 0.69% |
| Identity | 22.06% | 26.45% | 23.18% | 23.28% | 26.66% | 25.55% | 25.07% | 25.86% | 28.74% | 32.39% |
| Data Protection and Privacy | 1.80% | 1.87% | 2.37% | 3.13% | 4.20% | 4.88% | 4.74% | 5.90% | 7.01% | 7.91% |

Stages 4 to 6: The infrastructure progressively improves, including the Access Control List (ACL) (stage 4_4), Security Strategies (stage 5_5), Network Traffic Analysis (stage 4_8), and Cryptographic Applications (stage 5_8).

Stages 7 to 10: The infrastructure further expands to include higher-level security measures, such as Intrusion Detection Systems (stage 8_8), Cyberspace Situational Awareness (stage 8_4), and Secure Network Access Controls (stage 10_5).

(3) Identity and access management

Stages 1 to 3: Basic Identity Verification and Password Rules begin to be introduced (stage 2_4, stage 2_7, stage 3_4, stage 3_7).

Stages 4 to 6: Authentication Technologies are further developed, including Password-Based Authentication and more complex Identification Technologies (stage 4_3, stage 5_9).

Stages 7 to 10: Identity and Access Management begins to integrate with higher-level security measures such as Firewall Configuration and Secure Network Access Controls (stage 7_1, stage 8_1, stage 9_1).

(4) Data protection and privacy

Between Stage 3 and Stage 4: With the rise in cyberattacks, data protection and privacy begin to gain attention (stage 3_8, stage 4_5). From Stage 5 to Stage 7: Technologies such as encryption applications (stage 5_8, stage 7_4) and privacy protection (stage 6_5) start to be widely adopted. From Stage 8 to Stage 10: Data protection and privacy policies begin to merge with emerging technologies like ransomware and blockchain technology, creating a more comprehensive data security framework (stage 10_6, stage 10_7).

(5) Intrusion detection techniques

Stages 1 to 3: The basic concepts of intrusion detection begin to take shape (stage 1_8, stage 2_2, stage 3_1). Stages 4 to 6: Intrusion Detection Systems (IDS) gradually become an essential part of cybersecurity and start to integrate with Network Traffic Analysis (stage 4_2, stage 5_2, stage 6_1). Stages 7 to 10: Intrusion detection technology further matures, finding applications in a broader range of scenarios such as anomaly detection (stage 8_5), next-generation intrusion detection systems (stage 8_8), and detection of malware (stage 9_7).

(6) Traffic analysis technology

Stages 2 to 4: Network traffic analysis is introduced as a method of detection technologies for cyberattacks (stage 2_9, stage 4_8).

Stages 6 to 8: Traffic analysis technology progressively matures and begins to be applied in multiple areas such as Attack Detection (stage 6_1), Anomaly Detection (stage 8_2), and Intrusion Detection (stage 8_5).

Stages 9 to 10: Traffic analysis technology plays an increasingly important role in areas such as Cybercrime Analysis (stage 9_3) and Intrusion Detection (stage 10_2).

In summary, the developmental path in the field of cybersecurity is a continuously evolving and maturing process that involves the collaborative growth and application of multiple key technologies. The advancement of these technologies not only enhances the overall level of cybersecurity but also offers robust support for addressing the ever-changing cyber threats.

**3.1.3. Topic stability analysis.** Based on the topic feature analysis in Table 1, we have also found the presence of specific topics, such as cyberattacks, network traffic, and detection. These stable topics are shown in Table 3.

In the field of cybersecurity research, the persistent existence of cyberattacks, network traffic, and detection across different stages indicates a certain level of stability within the

**Table 3. Stable topics.**

| Topic | Stage-Topic |
|---|---|
| Cyber attack | stage 1_8 Detection of Cyberattacks |
| | stage 2_5 Analysis of cyberattacks |
| | stage 3_6 cyberattack |
| | stage 4_2 Detection of Cyberattacks |
| | stage 4_6 Analysis of Attack Graphs |
| | stage 5_1 Cyberattack |
| | stage 6_2 Identification of Cyberattacks |
| | stage 7_7 Cyberattacks and Defense Strategies |
| | stage 8_6 DOS Attack Consequences |
| | stage 10_4 Cyberattack Identification |
| Network Traffic | stage 2_9 Network Traffic Analysis |
| | stage 4_8 Policy-Based Network Traffic Analysis |
| | stage 5_2 Detection of Network Traffic |
| | stage 6_1 Detection of Network Traffic |
| | stage 8_2 Detection of Network Traffic |
| | stage 9_3 network traffic & cybercrime |
| | stage 10_2 Network Traffic Analysis in Intrusion Detection |
| Detection | stage 1_8 Detection of Cyberattacks |
| | stage 2_2 Detection of Cyberattacks |
| | stage 3_1 Detection Techniques |
| | stage 4_2 Detection of Cyberattacks |
| | stage 5_2 Detection of Network Traffic |
| | stage 6_1 Detection of Network Traffic |
| | stage 7_2 Detection of Botnets |
| | stage 8_2 Detection of Network Traffic |
| | stage 9_7 Detection of malware |
| | stage 10_2 Network Traffic Analysis in Intrusion Detection |

period of the study. This stability suggests that these topics have maintained a consistent level of attention throughout the temporal scope of the corpus, consistently remaining hot topics of research.

(1) The continuous evolution and persistent threat of cyber attack

Cyberattacks, as a direct threat to cybersecurity, have consistently been a central focus of research. The methods and forms of cyberattacks are continuously evolving, ranging from simple virus propagation to more complex forms such as ransomware, phishing attacks, and DDoS attacks. Each new type of attack presents fresh challenges for cybersecurity. Therefore, ongoing research into cyberattacks is critical to ensuring that cybersecurity defenses are continuously improved and remain effective against emerging threats.

(2) The crucial role of network traffic

Network traffic, as a stable topic in the research domain, highlights the importance of monitoring and analyzing network traffic for cybersecurity. Network traffic contains a wealth of information, including both regular communication data and potential characteristics of attack behaviors. Through real-time monitoring and in-depth analysis of network traffic, abnormal traffic patterns can be detected promptly, thus identifying potential cyberattacks.

Therefore, the continuous development and optimization of network traffic analysis techniques is significant for enhancing cybersecurity protection.

(3) The core of detection technology

The continued importance of detection is highlighted throughout the entire field of cybersecurity. It enables the timely identification of and response to cyber threats on demand. Whether it is the Intrusion Detection System (IDS), Security Information and Event Management (SIEM) system, or the more recent detection technologies based on artificial intelligence and machine learning, detection has always been a core component of cybersecurity defense strategies. With the advancement of attackers' techniques, it becomes particularly crucial to develop detection technologies that are capable of rapidly and accurately identifying new types of attacks.

In summary, the persistent presence and stability of cyberattacks, network traffic, and detection in cybersecurity research indicate their significant roles within the domain of network security. These enduring topics not only reveal the common challenges in the field of cybersecurity but also point to important directions for future research and technological development.

## 3.2. Refined evolution path analysis

In this study, we integrate topic modeling and word vector techniques to construct topic evolution paths by calculating inter-topic similarities and visualizing these paths using Sankey diagrams. Initially, we divide the time-series data into ten stages and apply the Latent Dirichlet Allocation (LDA) model to extract topics from the text data at each stage. By using the LDA model to the documents within each stage, we obtain the topic distributions for each period. After identifying the topics for each stage, we utilize word vector models such as Word2Vec to convert the keywords of each topic into word vectors. Subsequently, we generate vector representations of the topics through methods like weighted averaging. To measure the similarity between topics across different time windows, we calculate the cosine similarity between topic vectors of adjacent stages. This approach determines the continuity and evolutionary relationships of the topics. Based on the computed topic similarity matrix, we set an appropriate similarity threshold (e.g., 0.50) to filter topic pairs with similarities exceeding the threshold, thereby establishing associations between topics in adjacent time windows. These associations form the topic evolution paths, reflecting the development and changes in topics over time. As shown in Fig 4, the Sankey diagram constructed based on topic similarity demonstrates the evolution of topics in the field of network security from 2003 to 2022. The vertical flow in the Sankey diagram depicts the types and processes of topic evolution at each stage, while the horizontal blocks represent the distribution and significance of topics at each stage.

In thematic evolution analysis, the width of a thematic block represents the distribution proportion of that theme within a specific stage. As depicted in Fig 4, the widths of thematic blocks across various stages are generally similar, indicating a relatively balanced distribution of most themes throughout the stages. For instance, in stages 3 and 5, the majority of thematic blocks exhibit comparable widths, reflecting an equitable distribution of thematic significance. However, certain themes—such as stages 2_1, 2_3, 6_1, 7_7, 8_3, and 10_2—occupy smaller proportions in their respective stages, as detailed in Table 4. These themes received less attention during their corresponding periods, suggesting they were not focal points or core areas of research at that time. Notably, themes 6_1 and 10_2 lack sufficient depth in research, occupying relatively smaller proportions during specific periods, indicating a need for further development and exploration.

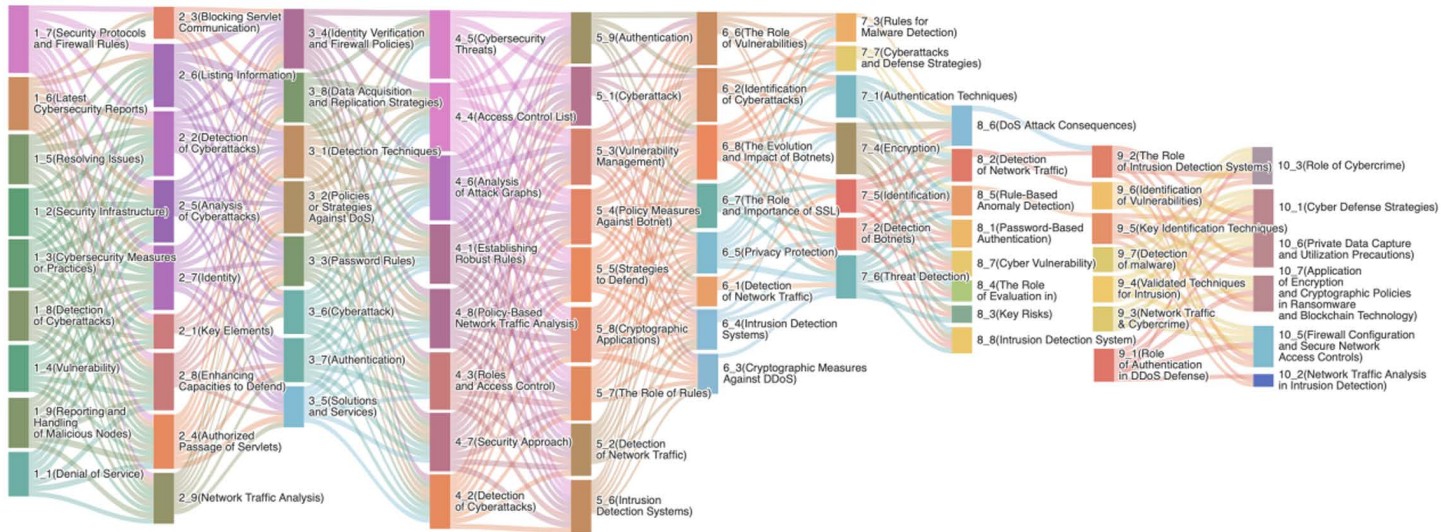

**Fig 4. Evolution paths of topics in cybersecurity.**

**Table 4. Topics-topic words.**

| Stages | Topics-Topic words |
| --- | --- |
| stage 2 | stage 2_1 Key Elements |
| stage 2 | stage 2_3 Blocking Servlet Communication |
| stage 6 | stage 6_1 Detection of Network Traffic |
| stage 7 | stage 7_7 Cyberattacks and Defense Strategies |
| stage 8 | stage 8_3 Key Risks |
| stage 10 | stage 10_2 Network Traffic Analysis in Intrusion Detection |

Additionally, the connecting lines between stages in Fig 4 illustrate the continuity of themes across adjacent stages. The density and uniform width of these lines between most stages suggest that the majority of themes exhibit strong continuity between successive stages. For example, from stage 5 to stage 6, nearly all themes are connected to the subsequent stage, demonstrating their sustained importance and continuity. Conversely, a few themes show diminished significance in certain stages, as evidenced by narrower thematic blocks and thinner connecting lines.

It can also be seen from Fig 4 that stages 1 to 6, 7 to 8, and 9 to 10 are more closely linked. Stages 1 to 6 cover the early development of cybersecurity technologies, such as the initial implementation of antivirus software, firewalls, and intrusion detection systems. During this period, cybersecurity topics centered on the establishment and optimization of foundational defense mechanisms. Stages 7 to 8 reflect a turning point in the field of cybersecurity, such as the PRISM incident in 2013, which sparked global attention to online privacy and surveillance, leading to a shift in security technology toward data and privacy protection. The emergence of APT attacks and ransomware during stages 9 to 10 indicates a further evolution in cybersecurity technology, with research focusing on countering more complex threats and ensuring data is not illegally encrypted or stolen. This has also led to more research focused on behavioral analysis and the application of artificial intelligence in security to better identify and respond to these advanced threats.

**3.2.1. Topic link feature analysis.** As shown in Fig 4, during stages 1 to 6, stages 7 to 8, and stages 9 to 10, there are multiple higher-weighted links between adjacent stages, indicating significant similarities. Here, we take the link feature from stage 4 to stage 5 as an example for analysis:

(1) Between stage 4 and stage 5, there are multiple higher-weighted links, suggesting a strong correlation between the two stages. This strong linkage represents a close continuity in terms of theory, methodology, or technology during the evolution from stage 4 to stage 5.

(2) The subsequent links of stage 4 cover 100%, and the preceding links of stage 5 also nearly cover 100%, showing that stage 5 has almost entirely evolved from the topics of stage 4. Every topic in Stage 5 can be revisited to multiple specific topics in Stage 4, and this one-to-many close connection demonstrates a highly segmented knowledge development process.

The sequential connectivity and high coverage between stages 4 and 5 reveal a distinct pattern of knowledge succession and topic progression. As Table 1 shows, the topics within Stage 4 encompassed key theories, methods, and technologies, which were further deepened, expanded, or applied upon entering Stage 5.

As depicted in Fig 4, there are fewer links between stages 6 and 7, as well as between stages 8 and 9, with lower weights and coverage rates compared to other adjacent stages, with stages 8 and 9 being more loosely linked.

As seen in Fig 4, stages 6 and 7, and stages 8 and 9 have fewer links compared to other adjacent stages, with lower weights and coverage rates, where the connections between stages 8 and 9 are even looser. Based on the features of the links from stage 6 to stage 7, the analysis is as follows:

(1) The lower link weights between multiple topics from stage 6 to stage 7 suggest that, although there is an information flow or topic evolution from stage 6 to stage 7, the influence of this flow or evolution is relatively weak. This means that the topics of stage 6 have a limited contribution to the development of stage 7, and the topic evolution of stage 7 is more significantly influenced by other factors.

(2) The subsequent link coverage from stage 6 is only 30%–60%, demonstrating that not all topics from stage 6 have had an impact on stage 7. This incomplete coverage reflects selective influences between topics, meaning that only a portion of stage 6 topics continues to play a role in stage 7, while other topics do not make a significant contribution.

(3) Although the preceding links of stage 7 also achieved only partial coverage, there are individual topics that realized full coverage in these links. This indicates that some topics in stage 7 have direct and significant connections with specific topics from stage 6. These links may point to topics or concepts in stage 6 that have a critical impact, which are then fully adopted or further developed in the evolution of stage 7.

This analysis reveals the complexity and selectivity of the transition from stage 6 to stage 7. While there are direct connections from stage 6 to stage 7, these links exhibit significant differences in strength and coverage. This reflects a non-linear process in the development of knowledge, the evolution of theory, or the application of practice. Within this process, certain topics or concepts are reinforced and expanded, while others may gradually fade from the focus of research or practice. Furthermore, it may also represent the development of topics in stage 7 that are newly influenced or based on a new knowledge base, rather than just a direct continuation of stage 6.

During the period from stage 6 (2013–2014) to stage 7 (2015–2016), attention to cybersecurity deepened both domestically and internationally, extending to the national security

level. The 2013 PRISM incident was a critical moment that signaled cybersecurity had become not just an issue of information technology but had risen to a focal point of national security concern. The surveillance program by the U.S. National Security Agency revealed large-scale monitoring of citizens' communication data by the government. Concurrently, this program also sparked widespread discussions and concerns about online privacy and government surveillance powers globally. This incident had a profound impact on the field of cybersecurity, advancing privacy-protecting technologies and accelerating the adoption of encrypted communication standards. In the academic sphere, research direction has made progress in traffic monitoring, attack identification, and security technologies. However, there was a weaker inheritance in research directions, which branched out into identity verification, encryption, identification tagging, and corresponding strategies.

The link analysis from stage 8 to stage 9 is as follows:

(1) These two stages have low linkage weights and coverage rates, with only three topics linking from stage 8 to stage 9. The weight of these links is relatively low, with a coverage rate of approximately 10%, indicating that the direct influence of stage 8 topics on stage 9 is very limited. This suggests that most topics in stage 8 did not significantly impact the development of stage 9, which likely emerged primarily under the influence of other factors or stages.

(2) The low coverage rate and the few topic links suggest a break or shift in topic evolution between stage 8 and stage 9. This generally occurs when there are significant changes in the research field, new technologies or theories are introduced, or major adjustments in research directions take place.

(3) Despite the limited influence from Stage 8 to Stage 9, the existing links still indicate a certain degree of knowledge or conceptual legacy.

(4) The formation of the topics in Stage 9 primarily relies on new topics or external influences that were not extensively discussed or explored in Stage 8. This suggests that new theories or technologies in cybersecurity research have emerged and that the hotspots and directions of research are undergoing rapid transformation.

The shift from Stage 8 to Stage 9 signifies that this is a complex process of topic evolution, which includes the partial inheritance of knowledge and concepts as well as notable changes in research directions or focal points.

During the period from stage 8 (2017–2018) to stage 9 (2019–2020), the massive outbreak of Advanced Persistent Threat (APT) attacks and ransomware shifted the focus within the cybersecurity field toward these more complex network threats. Simultaneously, the rapid development of machine learning technologies has provided new and effective research tools for the cybersecurity field, leading to the swift transformation of existing topics. In summary, technological breakthroughs and emerging security threats have driven a shift in focus and hot topics within the cybersecurity field, with researchers paying increasing attention to new topics and technologies.

### 3.2.2. Topic evolution type analysis.

(1) Merging type evolution

In this study, the merging evolution of topics exhibits significant characteristics of multiple merging, exerting a broad and profound influence on several topics in subsequent stages. Here, we choose only one topic evolution process to illustrate the merging evolution in this study. As shown in Fig 5, the formation and evolution of stage 2_4 reveal the intersection and merging of multiple topics, as well as the developmental trends of these merging topics.

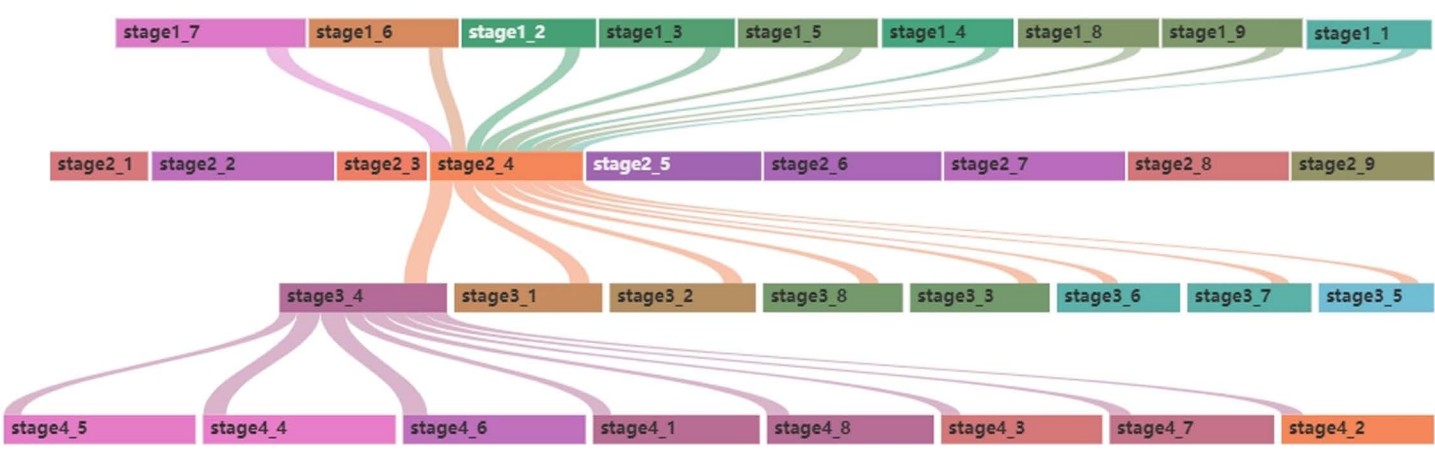

**Fig 5. Example of topic evolution types.**

The topics in Stage 1 represent the foundational areas of cybersecurity research, such as denial-of-service attacks, security infrastructure, network vulnerabilities, and the like [30,31]. By Stage 2, these foundational areas began to merge due to technological advancements and shifts in research trends, resulting in new topics that encompassed a broader range. Stage 2_4, namely Authorized Passage of Servlets, is the result of the merging of Java technology advancements and cybersecurity [32]. Java was indeed the focus of the industry during that period (2005, 2006).

As technology further advanced, although Servlet technology was gradually phased out by the industry, the related techniques and methods of stage 2_4 still exerted an influence on the subsequent development and evolution (in stage 3). This indicates that in the field of cybersecurity, certain topics, even if they no longer exist as independent research areas, can still have their influence merged into new research topics and directions.

(2) Inheritance type evolution

Through in-depth analysis of the evolution processes at each stage, it is found that there is no clear inheritance evolution pattern. The overall evolution process mostly demonstrates a dynamic relationship of many-to-many merging and division, rather than a single inheritance. Although there are a few links that show a trend of independent evolution, their coverage within the entire network remains relatively low. This finding emphasizes the importance of considering complex interactions in evolution path analysis and reveals the nonlinear characteristics of knowledge and technological development within this field.

(3) Division type evolution

Divisional evolution describes the process in which a topic develops over time into multiple, more refined subtopics. As shown in Fig 5, stage 3_4, or Identity Verification and Firewall Policies, is divided into multiple topics for stage 4. With in-depth research into identity recognition and firewall policies, stage 3_4 has influenced the development of research on attack detection, access control, network threat analysis, traffic analysis, and other topics in stage 4. This suggests that research in this field is expanding as researchers continue to explore additional subdivisions based on the combination of existing topics with others. Meanwhile, the emergence of new technologies and applications requires existing research topics to be updated or divided to adapt to these new findings.

(4) Generation type evolution

Fig 4 shows that topics stage 9_1, stage 9_3, stage 9_4, stage 9_7, and stage 10_4 are less influenced by preceding topics and are even entirely new research hotspots. The related topics list is as follows.

In Table 5, stage 9_4 and stage 10_4 represent a deepening of research on the previous topic, but they mainly incorporate new machine learning technologies, which have led to advancements in research and a regained focus within the industry [33,34]. Moreover, stages 9_1, 9_3, and 9_7 are entirely new research areas that have emerged with the continuous development of technology and applications [35–37].

(5) Disappearance type evolution

During the evolution process, stage 8_1, stage 8_3, stage 8_4, stage 8_6, and stage 8_8 no longer evolve as key topics, with the related topics listed as follows in Table 6.

These topics have a lesser impact on subsequent topics, reflecting the rapid development and evolution in the field of cybersecurity, where new technologies and strategies continuously complement or replace traditional approaches.

## 4. Discussions

In this paper, through the application of the LDA model, we successfully identified the core topics of each stage and summarized the topic features of each stage. Subsequently, we analyzed the evolution process of the cybersecurity field in terms of attack and defense technology progression, infrastructure maturity and expansion, refinement of identity and access management, data protection and privacy, intrusion detection techniques, and traffic analysis technologies. Furthermore, this study also observed the presence of topic stability phenomena within these topics during the research period. Finally, based on the density of links between stages in the Sankey diagram (Fig 4), combined with the analysis of topic characteristics, we examine the development process of cybersecurity technology.

**Table 5. Generation type topics.**

| Stages | Topics-Topic words |
|---|---|
| stage 9 | stage 9_1 Role of Authentication in DDoS Defense |
| | stage 9_3 Network Traffic & Cybercrime |
| | stage 9_4 Validated Techniques for Intrusion |
| | stage 9_7 Detection of Malware |
| stage 10 | stage 10_4 Cyberattack Identification |

**Table 6. Disappearance type topics.**

| Stages | Topics-Topic words |
|---|---|
| stage 8 | stage 8_1 Password-Based Authentication |
| | stage 8_3 Key Risks |
| | stage 8_4 The Role of Evaluation in |
| | stage 8_6 DOS Attack Consequences |
| | stage 8_8 Intrusion Detection System |

### 4.1. Topic evolution: From basic defense to forward-looking and comprehensive strategies

Through an in-depth analysis of the ten stages in this two-decade literature, we have witnessed significant transformations and advancements in the field of cybersecurity, as well as the continuous evolution and maturation of cybersecurity topics.

Between stages 1 and 3, the focus within the field of cybersecurity was primarily on infrastructure construction and routine security management. Enterprises and organizations began to recognize the importance of cybersecurity and gradually established a basic security protection system. However, with the increase in complications from cyberattacks, mere infrastructure was no longer sufficient to meet the demands. Therefore, in the subsequent stages (stages 4 to 6), the field of cybersecurity began to explore attack detection and defense technologies more deeply. Authentication, access control, and encryption technologies became the focus of research. At the same time, to cope with the ever-changing cyber threats, diversified defense strategies began to take shape.

Between stages 7 and 8, with the outbreak of the PRISM incident, academics began to focus on privacy protection and data security. With the gradual maturation of machine learning technologies (stages 8 to 9), along with the rapid development of emerging technologies such as cloud computing, the Internet of Things, 5G, and blockchain, the field of cybersecurity is again confronted with new challenges and opportunities. As the progressive maturation of machine learning technologies continues, emerging technologies such as cloud computing, the Internet of Things, 5G, and blockchain rapidly develop. The field of cybersecurity also faces new challenges and opportunities.

As a result, at stage 10, research begins to focus more on being forward-looking and innovative. Innovations in authentication and key technologies, intrusion detection, and authentication technologies, as well as the exploration of security applications in emerging technologies have become hot topics of research.

In conclusion, the evolution of topics in the field of cybersecurity has undergone a transformation from basic defense to forward-looking and comprehensive strategies. In the future, with the continuous advancement of technology and the ongoing evolution of threats, it is foreseeable that the field of cybersecurity will maintain a trend of innovation and development, making even greater contributions to the security and stability of the digital world.

### 4.2. Stage evolution: Dependencies in multi-stage interactions

The field of cybersecurity has experienced tremendous changes over the past two decades. From 2003 to 2022, the field went through multiple stages of evolution, each with its unique topics and developmental focuses. Through in-depth topic similarity analysis of these stages, we can observe the connections and evolving trends between each stage. (1) As shown in Fig 4, the Sankey diagram constructed based on topic similarity vividly demonstrates the process of topic evolution in the field of cybersecurity from 2003 to 2022. Among stages 1 to 6, 7 to 8, and 9 to 10, there are multiple high-weight links between adjacent stages. This signifies a high degree of topic similarity between neighboring stages, suggesting continuity and correlation in the topic development among these stages. It also demonstrates that research related to these topics is progressively deepening and broadening, resulting in an organic developmental path. More notably, the links achieve a coverage rate of 100% for their preceding and succeeding nodes, meaning that each stage is not only a continuation of the previous one but also lays the groundwork for the subsequent stages. Such a high coverage rate indicates that development in the field of cybersecurity is systematic, with each stage influencing and promoting the next. This phenomenon illustrates, for one thing, that with the continuous advancement of technology and the evolution of threats, cybersecurity issues are becoming increasingly complex

and diverse. To address these challenges, research within the field needs to deepen and expand continuously, forming a more comprehensive and systematic theoretical framework. For another thing, it suggests that researchers in the field of cybersecurity may have realized that only through systematic research and continuous accumulation can greater breakthroughs and innovations be achieved in this domain. (2) The links between Stage 6 and Stage 7, as well as between Stage 8 and Stage 9, show fewer connections with lower weights and coverage rates compared to other adjacent stages. This phenomenon reflects some significant changes or turning points in the development process of these stages in the field of cybersecurity. For instance, the 2013 PRISM incident prompted the academic community to shift focus toward privacy protection and data security, which indeed validates this observation. Moreover, the linkage between Stage 8 and Stage 9 appears particularly loose, revealing that during these adjacent stages, the field of cybersecurity underwent significant topic changes or shifts. Underlying this shift is the profound impact of the rapid development and practical application of machine learning technologies. When APT attacks and ransomware significantly impacted the industry, researchers were compelled to confront these new challenges and opportunities, leading to a reevaluation of current research topics and directions. Consequently, during this period, the topic development within the cybersecurity field underwent adjustment and reshaping to adapt to the new technological environment and threat situation. This also explains why the connection between stage 8 and stage 9 is relatively weaker, as they represent distinct stages in the evolution of cybersecurity in response to new challenges and opportunities.

## 4.3. Refinement: Topic evolution path analysis

By conducting stage topic analysis, we further performed a refined evolution paths analysis for the significant topic directions within the field of cybersecurity. (1) Attack and Defense Techniques. From the early days of basic defensive measures (stage 1_1, stage 2_1, stage 3_5), it evolved into the sophistication of attacks, and research began to focus on more advanced attack detection techniques (stage 4_6, stage 5_9). Ultimately, it delved into complex attack and defense strategies (stage 6_8, stage 7_7, stage 8_1, stage 9_1). (2) Infrastructures. They started with the initial establishment of basic cybersecurity infrastructure (stage 1_3, stage 1_5, stage 1_7) and evolved into progressively better infrastructure (stage 4_3, stage 5_5, stage 4_8, stage 5_8). Ultimately, the infrastructure further expanded, incorporating higher-level security measures (stage 8_8, stage 8_4, stage 7_10, stage 10_5). (3) Identity and Access Management. In the early stages, basic authentication and password policies were introduced (stage 2_4, stage 2_7, stage 3_4, stage 3_7). This was followed by further developments in identification technologies (stage 4_3, stage 5_9). Finally, as research deepened, related technologies were integrated (stage 7_1, stage 8_1, stage 9_1). (4) Data Protection and Privacy. With the rise in cyberattacks and the impact of cybersecurity incidents, data protection and privacy began to gain attention (stage 3_8, stage 4_5). This subsequently expanded to encryption applications (stage 5_8, stage 7_4) and privacy protection (stage 6_5). Ultimately, as research deepened, it extended to include data protection and privacy policy enforcement and ransomware (stage 10_6, stage 10_7). (5) Intrusion Detection Techniques. In the beginning stages, the basic concepts of intrusion detection began to take shape (stage 1_8, stage 2_2, stage 3_1). These concepts then started to be integrated with network traffic analysis (stage 4_2, stage 5_2, stage 6_1). Subsequently, they were applied to a broader range of scenarios, such as anomaly detection (stage 8_5), next-generation intrusion detection systems (stage 8_8), and malware detection (stage 9_7). (6) Traffic Analysis Technology. In the early stages, the community began to focus on traffic analysis techniques (stage 2_9). As these techniques matured (stage 4_8), they started to be applied in various areas, such as attack detection (stage 6_1) and anomaly detection (stage 8_2). Subsequently, the techniques evolved further into application stages, playing an increasingly important role in fields like intrusion detection (stage 8_5) and cybercrime analysis (stage 10_2).

### 4.4. Backtracking of topics: The dynamic balance between technological development and security threats

From Fig 4, we can clearly see that the development of topics within the field of cybersecurity is a dynamic process, with complex interrelationships between various stages. Among these stages, some topics exhibit a pronounced revisit phenomenon, meaning that these topics show a trend of initially declining and then rising over time. This indicates that in the process of topic development, some topics may experience a period of decline, but as technology advances and new threats emerge, these topics regain attention and are researched once again. This characteristic reflects the dynamic and balanced relationship between technological development and security threats. Through this analysis, we can gain a deeper understanding of the interaction and influence mechanisms between technological development and security threats. At the same time, it also offers valuable insights and guidance for research and practice in related fields. There are two reasons for this phenomenon. First, it is due to the complexity and dynamism inherent in the field of cybersecurity. With the continuous advancement of technology and the evolution of threats, new research directions and methods emerge, causing some early topics to be gradually replaced by new research hotspots. However, over time, some of the earlier issues may resurface as new threats or challenges and, therefore, need renewed research. The second reason is related to the changing trends in research and the shifting focus of researchers within the field. In the field of cybersecurity, researchers' focus may shift in response to emerging threats and challenges. When new threats emerge, researchers may devote more effort to countering these threats, which can result in some early research topics being temporarily overlooked. However, as new threats continue to evolve and change, researchers may revisit these earlier topics and find that some unresolved issues or challenges persist.

From a text-mining perspective, this phenomenon also reflects the characteristics of knowledge evolution in the field of cybersecurity. In text mining, the evolution patterns and trends of topics across various stages can be revealed through topic modeling and analysis of large volumes of textual data. The revisit phenomenon indicates that in the process of topic evolution, the significance and influence of some topics do not necessarily diminish over time. On the contrary, they may regain attention and be studied anew under new contexts. The topics exhibiting the revisit phenomenon are shown in Table 7. The proportion of literature in each stage is shown in Table 8.

## 5. Conclusions

This study employs the LDA2vc model and combines it with a phased approach to conduct an in-depth evolutionary analysis of cybersecurity research topics. It systematically reviews and analyzes the technological development path in cybersecurity, successfully outlining the context of topic development in this domain. The conclusions are as follows:

(1) In the analysis of topic evolution in the cybersecurity field, the LDA2vec model outperforms the DTM model. This study analyzes the thematic evolution in the field of cybersecurity and reveals that, although the DTM demonstrates relatively better topic coherence across most stages (for example, in Stage 1, the topic coherence score of the LDA model

**Table 7. Topic revisit matrix.**

| Original Topic | Revisited Topic words |
|---|---|
| stage 1_1 | stage 8_6 |
| stage 1_7 | stage 10_5 |

**Table 8. Proportion of literature by topic revisited.**

| Topics | stage 1 | stage 2 | stage 3 | stage 4 | stage 5 | stage 6 | stage 7 | stage 8 | stage 9 | stage 10 |
|---|---|---|---|---|---|---|---|---|---|---|
| stage 1_1 stage 8_6 | 8.08% | 5.55% | 5.67% | 4.41% | 3.88% | 4.83% | 4.02% | 3.97% | 3.88% | 4.76% |
| stage 1_4 stage 8_7 | 0.70% | 0.81% | 0.71% | 0.97% | 0.92% | 1.27% | 1.17% | 1.04% | 1.08% | 1.14% |

is −11.8 while that of the DTM is −12.2), and its score gradually increases over time, the limitations of the DTM in semantic recognition depth result in weaknesses in topic word identification (as shown in Fig 2). This indicates that the DTM has certain limitations in the accuracy and effectiveness of topic evolution paths. In contrast, the LDA2Vec model exhibits a more significant advantage in both topic differentiation and evolution path construction, effectively revealing the relationships and evolutionary trends between topics across different stages.

(2) The research topics in the field of cybersecurity exhibit characteristics of complexity and diversity. Through an in-depth analysis of the cybersecurity literature from 2003 to 2022, this study extracted 35 distinct topics that show significant trends of change over time. The proportion of literature related to each topic demonstrates noticeable fluctuations across different stages. For instance, the "Attack and Defense" topics maintained a high proportion of literature throughout all stages, increasing from 37.62% in Stage 1 to 50.52% in Stage 10. In contrast, the "Intrusion Detection" topics, while exhibiting an overall decreasing trend, demonstrate volatility: the proportion dropped from 23.35% in Stage 1 to 9.81% in Stage 8 but then rebounded to 12.39% in Stage 10. This fluctuation indicates that although "Intrusion Detection" was relatively less emphasized during certain stages, emerging security threats led to renewed attention in subsequent years. Particularly in later stages, the need for intrusion detection technologies to address novel attack methods resulted in a resurgence of research in this field. Furthermore, the proportions of literature related to "Traffic Analysis" and "Infrastructures" showed more stability but exhibited downward trends. Notably, "Identity" topics steadily increased from 22.06% in Stage 1 to 32.39% in Stage 10, reflecting ongoing interest and deeper research in this area. The "Data Protection and Privacy" topic, which became a focal point in 2007, saw its literature proportion rise from 1.80% to 7.91%, particularly driven by advancements in big data, cloud computing, and artificial intelligence, further intensifying research in this field.

(3) There is a dynamic and balanced relationship between technological development and security threats in the cybersecurity domain. With the continuous advancement of technology, certain research topics have experienced a noticeable reversal, indicating that early security issues have regained attention driven by new technologies. For instance, the topic "1_1 (Denial of Service) and 8_6 (DoS Attack Consequences)" accounted for 8.08% of the literature in Stage 1 but gradually declined to 4.02% by Stage 7. However, in the later stages, this topic rebounded to 4.76% in Stage 10, suggesting that with the application of new technologies and the emergence of novel attack methods, DoS attacks and their consequences have once again become a focal point of research. Similarly, the topic "1_4 (Vulnerability) and 8_7 (Vulnerability)" had a literature proportion of 0.70% in Stage 1 and fluctuated in subsequent stages, reaching a peak of 1.27% in Stage 6 before experiencing a decline. However, starting from Stage 8, the proportion of this topic increased again, reaching 1.14% in Stage 10. This phenomenon demonstrates that, although some early security issues were neglected during technological progress, the emergence of new

vulnerabilities has led to these issues regaining their focus in research. This reversal phenomenon reflects the sensitivity of the cybersecurity research field to the ever-changing threat landscape, as well as the interplay between technological innovation and security demands. Driven by new technologies, previously overlooked security issues have once again attracted attention, highlighting the intertwined relationship between technological advancement and security threats.

The research contributions of this paper are as follows:

(1) This study fully leverages the advantages of the LDA model in topic extraction and combines it with a phased division strategy. The analysis results effectively reflect the evolutionary patterns of research topics in the field of cybersecurity.

(2) This study identifies the key evolutionary stages in the field of cybersecurity, revealing the development trajectory from basic defense mechanisms to complex threat detection and response strategies. Specifically, the focus from 2003 to 2008 was on network infrastructure development and defense strategies, followed by a focus on intrusion detection and identity management from 2009 to 2016. From 2017 to 2022, attention shifted toward privacy protection and more advanced attack detection technologies.

(3) This study analyzes the development trajectory of topics in the field of cybersecurity from various aspects, including the evolution of attack and defense technologies, infrastructure maturity and expansion, refinement of identity and access management, data protection and privacy, intrusion detection techniques, and traffic analysis technologies. Additionally, while analyzing the evolutionary path, this research also examines phenomena such as merging, inheritance, division, generation, and disappearance based on Sankey diagrams. These findings reflect the changing research hotspots in the field of cybersecurity, providing not only a systematic perspective on literature evolution for cybersecurity research but also revealing shifts in research focus and trends.

This study has the following limitations:

(1) The depth of methodological integration needs to be strengthened. Although the LDA-2vec model combines the advantages of the LDA and Word2vec models, the research lacks sufficient exploration of optimizing and adjusting model parameters.

(2) The evaluation criteria for the model should be more diverse. The current study conducts only a comparative analysis using the DTM model and primarily focuses on topic words and coherence indicators, leading to a singular set of evaluation criteria.

In future research, the research team will build upon the current study by using the latest datasets to explore the thematic evolution analysis in cybersecurity further. Specifically, future studies will optimize the LDA2Vec model by incorporating deep learning models, such as BERT and GPT, to enhance its ability to identify long texts and interdisciplinary topics. Additionally, we plan to introduce more cross-platform datasets (e.g., IEEE, Scopus) for comparative analysis to validate the generalizability and effectiveness of the methodology in different academic environments. The current study primarily conducts a comparative analysis using DTM; however, future work will consider incorporating more dimensional evaluation metrics, such as topic distinguishability and topic stability, to assess the performance of different topic models across stages more comprehensively. Through these improvements, we aim to provide richer data support and deeper insights, further aiding the understanding of the knowledge structure and development trends in the field of cybersecurity, thereby fostering continued progress and innovation in this area of research.

## Author contributions

**Conceptualization:** Yanfeng Zhu.

**Data curation:** Yanfeng Zhu, Zheng Li.

**Funding acquisition:** Lei Jiang.

**Methodology:** Lei Jiang.

**Visualization:** Zheng Li.

**Writing – original draft:** Yanfeng Zhu, Zheng Li.

**Writing – review & editing:** Tianyi Li.

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
