## [Decision Letter · Decision Letter 0]

4 Dec 2024

PONE-D-24-13562Topic Recognition and Refined Evolution Path Sustainability Analysis of Literature in the Field of CybersecurityPLOS ONE

Dear Dr. Li,

Thank you for submitting your manuscript to PLOS ONE. After careful consideration, we feel that it has merit but does not fully meet PLOS ONE’s publication criteria as it currently stands. Therefore, we invite you to submit a revised version of the manuscript that addresses the points raised during the review process.

We look forward to receiving your revised manuscript.

Kind regards,

Burak Erkayman

Academic Editor

PLOS ONE

Journal Requirements: When submitting your revision, we need you to address these additional requirements. 1. Please ensure that your manuscript meets PLOS ONE's style requirements, including those for file naming. The PLOS ONE style templates can be found at https://journals.plos.org/plosone/s/file?id=wjVg/PLOSOne_formatting_sample_main_body.pdf and https://journals.plos.org/plosone/s/file?id=ba62/PLOSOne_formatting_sample_title_authors_affiliations.pdf 2. Please note that PLOS ONE has specific guidelines on code sharing for submissions in which author-generated code underpins the findings in the manuscript. In these cases, all author-generated code must be made available without restrictions upon publication of the work. Please review our guidelines at https://journals.plos.org/plosone/s/materials-and-software-sharing#loc-sharing-code and ensure that your code is shared in a way that follows best practice and facilitates reproducibility and reuse 3. We note that the grant information you provided in the ‘Funding Information’ and ‘Financial Disclosure’ sections do not match.  When you resubmit, please ensure that you provide the correct grant numbers for the awards you received for your study in the ‘Funding Information’ section. 4. Thank you for stating the following financial disclosure: "This study was supported by the Project of Social Science Foundation of Heilongjiang Province (No. 23TQD174), and the Project of Basic Research Operating  Costs of Provincial Higher Education Institutions in Heilongjiang Province (No. 2022-KYYWF-1052)." Please state what role the funders took in the study.  If the funders had no role, please state: ""The funders had no role in study design, data collection and analysis, decision to publish, or preparation of the manuscript."" If this statement is not correct you must amend it as needed. Please include this amended Role of Funder statement in your cover letter; we will change the online submission form on your behalf. 5. We note that your Data Availability Statement is currently as follows: All relevant data are within the manuscript and its Supporting Information files. Please confirm at this time whether or not your submission contains all raw data required to replicate the results of your study. Authors must share the “minimal data set” for their submission. PLOS defines the minimal data set to consist of the data required to replicate all study findings reported in the article, as well as related metadata and methods (https://journals.plos.org/plosone/s/data-availability#loc-minimal-data-set-definition). For example, authors should submit the following data: - The values behind the means, standard deviations and other measures reported;- The values used to build graphs;- The points extracted from images for analysis. Authors do not need to submit their entire data set if only a portion of the data was used in the reported study. If your submission does not contain these data, please either upload them as Supporting Information files or deposit them to a stable, public repository and provide us with the relevant URLs, DOIs, or accession numbers. For a list of recommended repositories, please see https://journals.plos.org/plosone/s/recommended-repositories. If there are ethical or legal restrictions on sharing a de-identified data set, please explain them in detail (e.g., data contain potentially sensitive information, data are owned by a third-party organization, etc.) and who has imposed them (e.g., an ethics committee). Please also provide contact information for a data access committee, ethics committee, or other institutional body to which data requests may be sent. If data are owned by a third party, please indicate how others may request data access. 6. PLOS requires an ORCID iD for the corresponding author in Editorial Manager on papers submitted after December 6th, 2016. Please ensure that you have an ORCID iD and that it is validated in Editorial Manager. To do this, go to ‘Update my Information’ (in the upper left-hand corner of the main menu), and click on the Fetch/Validate link next to the ORCID field. This will take you to the ORCID site and allow you to create a new iD or authenticate a pre-existing iD in Editorial Manager. 7. We note you have included a table to which you do not refer in the text of your manuscript. Please ensure that you refer to Table 5 in your text; if accepted, production will need this reference to link the reader to the Table.

Reviewers' comments:

Reviewer's Responses to Questions

**Comments to the Author**

1. Is the manuscript technically sound, and do the data support the conclusions?

Reviewer #1: Partly

Reviewer #2: Yes

2. Has the statistical analysis been performed appropriately and rigorously? 

Reviewer #1: No

Reviewer #2: Yes

3. Have the authors made all data underlying the findings in their manuscript fully available?

Reviewer #1: No

Reviewer #2: Yes

4. Is the manuscript presented in an intelligible fashion and written in standard English?

Reviewer #1: No

Reviewer #2: Yes

5. Review Comments to the Author

Reviewer #1: The topic is interesting.

The amount of literature surveyeed is fine. The ideaa of tracing the evolution of the literature is good.

The choice of topic modelling techniques is fine.

The use the authors make of LDA is quite questionable. LDA provides a list of words for each topic with their probability but do not provide labels for the topics. Where do the topic words in Table 1 come from? All the subsequent analysis is based on those words. If those labels have been assigned by the authors, a degree of subjectivity is present and may harm the validity of the results.

The analysis of the evolution is fuzzy at its best. Figure 4 is unreadable and appears as a mess of flows. The authors should provide a better explanation of this kind of representation.

Also, the evolution analysis of topics should be examined through a quantitative framework, for example using the Jaccard index. Also, it may be neagtively impacted by the partial overlapping of some topic keywords (for example, what is the difference between "Detection of network traffic" and "Network traffic detection" in Table 2?).

The paper should be carefully revised for English, as there are many typos (e.g., txet instead of text in Figure 1) and wrong expressions (e.g., "this study divided of literature into ten stages" at the end of Section 3.1).

Reviewer #2: 1- Why is the year 2023 not covered in this research review at least?

2- The introduction is very simple and is not presented in a gradual manner from the general work environment to the specific research area that this manuscript is concerned with.

3- The research papers in the related works are not organized in order according to the year of publication (from oldest to newest). What are the contributions of the researcher through this manuscript compared to them?

4- In Figure 1 the texts need to be enlarged to be clearer.

5- The research papers used are very few and the researcher did not provide any comparison of the advantages and disadvantages of each method presented to clarify the vision.

6- Use numerical values for results in conclusions and to explain proposed future work.

7- Ensure that all data for references is placed and their format is standardized

6. PLOS authors have the option to publish the peer review history of their article (what does this mean? ). If published, this will include your full peer review and any attached files.

**Do you want your identity to be public for this peer review?** For information about this choice, including consent withdrawal, please see our Privacy Policy .

Reviewer #1: No

Reviewer #2: No

---

## [Author Response · Author response to Decision Letter 1]

25 Dec 2024

Description of the Revision

Dear Editor:

We are very grateful for your constructive comments and suggestions for our manuscript (NO. PONE-D-24-13562). Due to the extensive nature of the revisions, we have highlighted the main modifications in red [to be reflected in the submitted paper]. The specific explanations for these changes are as follows:

Response to Reviewer 1 Comments

Point 1: The use the authors make of LDA is quite questionable. LDA provides a list of words for each topic with their probability but does not provide labels for the topics.

Where do the topic words in Table 1 come from? All the subsequent analysis is based on those words. If those labels have been assigned by the authors, a degree of subjectivity is present and may harm the validity of the results. The analysis of the evolution is fuzzy at its best.

Response 1: Thank you for pointing this out. We have carefully revised this comment. The revised section is highlighted in red. The modifications are as follows:

To identify the core research themes and their evolution across different stages in the field of cybersecurity, this study employed the LDA model for training and topic extraction from the literature. The LDA model analyzes a collection of documents to identify multiple topics, each represented by a set of words and their associated probability distribution. However, LDA does not automatically generate labels for these topics, requiring manual assignment based on the semantic meanings of the high-probability terms within each topic. For instance, in Topic 2 of Stage 2, the high-probability words and their probabilities are as follows: detection (0.073), cyberattack (0.050), cybersecurity_threats (0.033), forms (0.029), survive (0.023), cybersecurity (0.015), identification (0.015), spurious (0.012), avoid (0.011), and protecting (0.011). Based on the semantics of these high-probability words, the topic can be labeled as "Stage2_2: Detection of Cyberattacks." The topic words listed in Table 1 were generated following this procedure.

Additionally, the number of topics for each stage was determined by calculating the perplexity for each stage. To avoid overfitting, the value corresponding to the inflection point of the perplexity curve was chosen as the optimal number of topics. Using this method, the optimal number of topics for the ten stages was determined to be 9, 9, 8, 8, 9, 8, 7, 8, 7, and 7, respectively.

Point 2: Figure 4 is unreadable and appears as a mess of flows. The authors should provide a better explanation of this kind of representation.

Response 2: Thank you for pointing out the readability issue with Figure 4. We have redesigned the figure with the following improvements:

Enhancement of Graphic Clarity: We have redrawn Figure 4 to ensure its resolution meets the journal's requirements, thereby improving image clarity.

Layout Optimization: The arrangement of elements within the figure has been adjusted to present information more orderly, avoiding the previous clutter.

Colors and Labels: More intuitive colors and labeling schemes have been adopted to enable readers to distinguish different processes and data more easily.

Subject Annotation: Specific subject titles have been added within the figure to assist readers in understanding the content presented in Figure 4.

The revised Figure 4 is as follows:

Point 3: Also, the evolution analysis of topics should be examined through a quantitative framework, for example using the Jaccard index.

Response 3: Thank you for pointing this out. We have carefully revised this comment. The revised section is highlighted in red. The modifications are as follows:

In this study, we integrate topic modeling and word vector techniques to construct topic evolution paths by calculating inter-topic similarities and visualizing these paths using Sankey diagrams. Initially, we divide the time-series data into ten stages and apply the Latent Dirichlet Allocation (LDA) model to extract topics from the text data at each stage. By using the LDA model to the documents within each stage, we obtain the topic distributions for each period. After identifying the topics for each stage, we utilize word vector models such as Word2Vec to convert the keywords of each topic into word vectors. Subsequently, we generate vector representations of the topics through methods like weighted averaging.

To measure the similarity between topics across different time windows, we calculate the cosine similarity between topic vectors of adjacent stages. This approach determines the continuity and evolutionary relationships of the topics. Based on the computed topic similarity matrix, we set an appropriate similarity threshold (e.g., 0.50) to filter topic pairs with similarities exceeding the threshold, thereby establishing associations between topics in adjacent time windows. These associations form the topic evolution paths, reflecting the development and changes in topics over time, as illustrated in Figure 4.

Point 4: Also, it may be negatively impacted by the partial overlapping of some topic keywords (for example, what is the difference between "Detection of network traffic" and "Network traffic detection" in Table 2?).

Response 4: We are grateful for the reviewer's comments. During the thematic refinement process, we observed that although these two themes share the highest-frequency core terms, they exhibit significant differences in the composition of other high-frequency terms. Given this important finding, we carefully decided to distill two distinct themes to ensure their precision and representativeness. A similar situation was encountered in handling themes 5_2, 6_1, and 8_2. After a thorough review and incorporation of valuable feedback from experts, we recognized that our initial interpretation of certain aspects may have inadvertently led to a negative impact on the themes' evolutionary trajectory. In response, we conducted a rigorous reexamination and made necessary revisions, aiming to present the content in a more objective and balanced manner while avoiding any potential misunderstandings. This adjustment not only enhanced the accuracy of our analysis but also further strengthened the scientific rigor and credibility of the report.

Due to the extensive nature of the revisions, the specific changes are not detailed here. We have highlighted the main modifications in red in the submitted manuscript.

Point 4: The paper should be carefully revised for English, as there are many typos (e.g., txet instead of text in Figure 1) and wrong expressions (e.g., "this study divided of literature into ten stages" at the end of Section 3.1).

Response 4: We are grateful for the reviewer's comments. We have carefully reviewed the entire manuscript and revised it to correct all identified errors. Due to the extensive nature of the revisions, the specific changes are not detailed here. We have highlighted the main modifications in red within the submitted manuscript. Figure 1 has been revised. The incorrect expression "this study divided of literature into ten stages" has been revised to "this study divided the documents into ten stages" at the end of Section 3.1.

Response to Reviewer 2 Comments

Point 5: Why is the year 2023 not covered in this research review at least?

Response 5: Thank you for pointing this out. We have carefully revised this comment. The explanation for this problem is as follows:

This study covers the data collection period from 2003 to 2022. Due to the incomplete availability of data for the year 2023 at the time of analysis and manuscript preparation, we selected 2022 as the cutoff year for the data. This decision was made to ensure the integrity of the data and the accuracy of the analysis, thereby preventing potential biases or inaccuracies that could arise from incomplete data.

As discussed in the outlook section of the paper, we plan to continue the investigation into the evolution of themes in the field of cybersecurity in future research. Subsequent analyses will include data from 2023 and beyond. This ongoing data collection and analysis will allow us to track the latest developments in the research themes and verify the continued applicability of the conclusions drawn in the current study.

Point 6: The introduction is very simple and does not gradually transition from the general work environment to the specific research area that this manuscript concerns.

Response 6: We are grateful for the reviewer's comments. Due to the extensive nature of the revisions, the specific changes are not detailed here. We have highlighted the main modifications in red within the submitted manuscript.

Point 7: The research papers in the related works are not organized in order according to the year of publication (from oldest to newest). What are the contributions of the researcher through this manuscript compared to them?

Response 7: We are grateful for the reviewer's comments. Due to the extensive nature of the revisions, the specific changes are not detailed here. We have highlighted the main modifications in red within the submitted manuscript.

Point 8: In Figure 1 the texts need to be enlarged to be clearer.

Response 8: We are grateful for the reviewer's comments. Figure 1 has been revised as follows:

Point 9: The research papers used are very few and the researcher did not provide any comparison of the advantages and disadvantages of each method presented to clarify the vision.

Response 9: Thank you for pointing this out. We have carefully revised this comment. Due to the extensive nature of the revisions, the specific changes are not detailed here. We have highlighted the main modifications in red within the submitted manuscript.

Point 10: Use numerical values for results in conclusions and to explain proposed future work.

Response 10: We are grateful for the reviewer's comments. After careful consideration, we have added content about the number of topics in the form of a table and modified the conclusion. All revised sections are highlighted in red in the submitted manuscript. The relevant tables and conclusion sections are as follows:

This study employs the LDA2vc model and combines it with a phased approach to conduct an in-depth evolutionary analysis of cybersecurity research topics. It systematically reviews and analyzes the technological development path in cybersecurity, successfully outlining the context of topic development in this domain. The conclusions are as follows:

(1) In the analysis of topic evolution in the cybersecurity field, the LDA2vec model outperforms the DTM model. This study analyzes the thematic evolution in the field of cybersecurity and reveals that, although the DTM demonstrates relatively better topic coherence across most stages (for example, in Stage 1, the topic coherence score of the LDA model is -11.8 while that of the DTM is -12.2), and its score gradually increases over time, the limitations of the DTM in semantic recognition depth result in weaknesses in topic word identification (as shown in Figure 2). This indicates that the DTM has certain limitations in the accuracy and effectiveness of topic evolution paths. In contrast, the LDA2Vec model exhibits a more significant advantage in both topic differentiation and evolution path construction, effectively revealing the relationships and evolutionary trends between topics across different stages.

(2) The research topics in the field of cybersecurity exhibit characteristics of complexity and diversity. Through an in-depth analysis of the cybersecurity literature from 2003 to 2022, this study extracted 35 distinct topics that show significant trends of change over time. The proportion of literature related to each topic demonstrates noticeable fluctuations across different stages. For instance, the "Attack and Defense" topics maintained a high proportion of literature throughout all stages, increasing from 37.62% in Stage 1 to 50.52% in Stage 10. In contrast, the "Intrusion Detection" topics, while exhibiting an overall decreasing trend, demonstrate volatility: the proportion dropped from 23.35% in Stage 1 to 9.81% in Stage 8 but then rebounded to 12.39% in Stage 10. This fluctuation indicates that although "Intrusion Detection" was relatively less emphasized during certain stages, emerging security threats led to renewed attention in subsequent years. Particularly in later stages, the need for intrusion detection technologies to address novel attack methods resulted in a resurgence of research in this field. Furthermore, the proportions of literature related to "Traffic Analysis" and "Infrastructures" showed more stability but exhibited downward trends. Notably, "Identity" topics steadily increased from 22.06% in Stage 1 to 32.39% in Stage 10, reflecting ongoing interest and deeper research in this area. The "Data Protection and Privacy" topic, which became a focal point in 2007, saw its literature proportion rise from 1.80% to 7.91%, particularly driven by advancements in big data, cloud computing, and artificial intelligence, further intensifying research in this field.

(3) There is a dynamic and balanced relationship between technological development and security threats in the cybersecurity domain. With the continuous advancement of technology, certain research topics have experienced a noticeable reversal, indicating that early security issues have regained attention driven by new technologies. For instance, the topic "1_1 (Denial of Service) and 8_6 (DoS Attack Consequences)" accounted for 8.08% of the literature in Stage 1 but gradually declined to 4.02% by Stage 7. However, in the later stages, this topic rebounded to 4.76% in Stage 10, suggesting that with the application of new technologies and the emergence of novel attack methods, DoS attacks and their consequences have once again become a focal point of research. Similarly, the topic "1_4 (vulnerability) and 8_7 (Vulnerability)" had a literature proportion of 0.70% in Stage 1 and fluctuated in subsequent stages, reaching a peak of 1.27% in Stage 6 before experiencing a decline. However, starting from Stage 8, the proportion of this topic increased again, reaching 1.14% in Stage 10. This phenomenon demonstrates that, although some early security issues were neglected during technological progress, the emergence of new vulnerabilities has led to these issues regaining their focus in research. This reversal phenomenon reflects the sensitivity of the cybersecurity research field to the ever-changing threat landscape, as well as the interplay between technological innovation and security demands. Driven by new technologies, previously overlooked security issues have once again attracted attention, highlighting the intertwined relationship between technological advancement and security threats.

The research contributions of this paper are as follows:

(1) This study fully leverages the advantages of the LDA model in topic extraction and combines it with a phased division strategy. The analysis results effectively reflect the evolutionary patterns of research topics in the field of cybersecurity.

(2) This study identifies the key evolutionary stages in the field of cybersecurity, revealing the development trajectory from basic defense mechanisms to complex threat detection and response strategies. Specifically, the focus from 2003 to 2008 was on network infrastructure development and defense strategies, followed by a focus on intrusion detection and identity management from 2009 to 2016. From 2017 to 2022, attention shifted toward privacy protection and more advanced attack detection technologies.

(3) This study analyzes the development trajectory of topics in the field of cybersecurity from various aspects, including the evolution of attack and defense technologies, infrastructure maturity and expansion, refinement of identity and access management, data protection and privacy, intrusion detection techniques, and traffic analysis technologies. Additionally, while analyzing the evolutionary path, this research also examines phenomena such as merging, inheritance, division, generation, and disappearance based on Sankey diagrams. These findings reflect the changing research hotspots in the field of cybersecurity, providing not only a systematic persp

---

## [Decision Letter · Decision Letter 1]

29 Jan 2025

Topic Recognition and Refined Evolution Path Analysis of Literature in the Field of Cybersecurity

PONE-D-24-13562R1

Dear Dr. Li,

We’re pleased to inform you that your manuscript has been judged scientifically suitable for publication and will be formally accepted for publication once it meets all outstanding technical requirements.

Kind regards,

Burak Erkayman

Academic Editor

PLOS ONE

Additional Editor Comments (optional):

Reviewers' comments:

Reviewer's Responses to Questions

**Comments to the Author**

1. If the authors have adequately addressed your comments raised in a previous round of review and you feel that this manuscript is now acceptable for publication, you may indicate that here to bypass the “Comments to the Author” section, enter your conflict of interest statement in the “Confidential to Editor” section, and submit your "Accept" recommendation.

Reviewer #2: All comments have been addressed

2. Is the manuscript technically sound, and do the data support the conclusions?

Reviewer #2: (No Response)

3. Has the statistical analysis been performed appropriately and rigorously? 

Reviewer #2: (No Response)

4. Have the authors made all data underlying the findings in their manuscript fully available?

Reviewer #2: (No Response)

5. Is the manuscript presented in an intelligible fashion and written in standard English?

Reviewer #2: (No Response)

6. Review Comments to the Author

Reviewer #2: (No Response)

7. PLOS authors have the option to publish the peer review history of their article (what does this mean? ). If published, this will include your full peer review and any attached files.

**Do you want your identity to be public for this peer review?** For information about this choice, including consent withdrawal, please see our Privacy Policy .

Reviewer #2: No

---

## [Editor Report · Acceptance letter]

PONE-D-24-13562R1

PLOS ONE

Dear Dr. Li,

I'm pleased to inform you that your manuscript has been deemed suitable for publication in PLOS ONE. Congratulations! Your manuscript is now being handed over to our production team.

Kind regards,

on behalf of

Dr. Burak Erkayman

Academic Editor

PLOS ONE